# Transcriptional Regulation of the Hippo Pathway: Current Understanding and Insights from Single-Cell Technologies

**DOI:** 10.3390/cells11142225

**Published:** 2022-07-17

**Authors:** Sayantanee Paul, Shiqi Xie, Xiaosai Yao, Anwesha Dey

**Affiliations:** 1Department of Discovery Oncology, Genentech Inc., 1 DNA Way, South San Francisco, CA 94080, USA; paul.sayantanee@gene.com (S.P.); xie.shiqi@gene.com (S.X.); 2Department of Oncology Bioinformatics, Genentech Inc., 1 DNA Way, South San Francisco, CA 94080, USA

**Keywords:** Hippo signaling, TEAD, transcriptional regulation, single cell technologies

## Abstract

The Hippo pathway regulates tissue homeostasis in normal development and drives oncogenic processes. In this review, we extensively discuss how YAP/TAZ/TEAD cooperate with other master transcription factors and epigenetic cofactors to orchestrate a broad spectrum of transcriptional responses. Even though these responses are often context- and lineage-specific, we do not have a good understanding of how such precise and specific transcriptional control is achieved—whether they are driven by differences in TEAD paralogs, or recruitment of cofactors to tissue-specific enhancers. We believe that emerging single-cell technologies would enable a granular understanding of how the Hippo pathway influences cell fate and drives oncogenic processes, ultimately allowing us to design better pharmacological agents against TEADs and identify robust pharmacodynamics markers of Hippo pathway inhibition.

## 1. Introduction

Hippo signaling pathway is a key modulator of tissue growth with widespread implications in organ development, cell growth, regeneration, and stem cell function [1]. It is at the crossroads of several upstream signaling events that control the activation and deactivation of Yes-associated protein (YAP) and transcriptional coactivator with PDZ-binding motif (TAZ)—two homologous proteins, collectively known as YAP/TAZ (*Drosophila* ortholog Yorkie, Yki) (Figure 1). YAP/TAZ transcription complex, when in the nucleus, associates with multiple transcription factors and activates gene networks that signal cell proliferation and survival. Conversely, sequestration of the YAP/TAZ complex in the cytosol is critical to finetune cell fate regulation. YAP/TAZ is phosphorylated by members of the NDR family kinases, Large Tumor Suppressor 1/2 (LATS1/2; *Drosophila* ortholog Warts, Wts). LATS1/2 is regulated by kinase, Mammalian STE20-like 1/2 (MST1/2; *Drosophila* homolog Hippo, Hpo), and their adaptor proteins SAV1 (*Drosophila* homolog Salvador, Sav) and MOB1 (*Drosophila* homolog Mats) [2]. Phosphorylation of the YAP/TAZ complex by LATS1/2 leads to destabilization of the complex and renders it inaccessible to the nucleus via promoting interaction with cytoplasmic protein 14-3-3. Nuclear localization of YAP/TAZ is critical for its function as transcriptional co-activators [2,3].

YAP/TAZ lack a DNA-binding domain, and primarily rely on their interaction with additional transcription factors (TFs) to exert transcriptional activity [8,9] (Figure 2). YAP/TAZ can also recruit co-factors and chromatin remodelers, forming distinct transcriptional modules that bind to cis-regulatory elements and govern transcriptional output. This adds layers of regulation to the transcriptional machinery, allowing it to receive information from several upstream signaling events, as well as to finetune gene expression on demand. Conversely, deregulation of YAP/TAZ or its associated factors leads to aberrant gene expression in cancer. YAP/TAZ has also recently emerged as a critical nexus contributing to resistance mechanisms against therapeutic interventions [10,11]. Although most of the YAP-associated gene expression networks require association of the TEAD family of transcription factors (TEAD 1-4) (*Drosophila* ortholog Scalloped, Sd), TEAD-independent gene regulation networks have also been documented [1].

Tumor heterogeneity, functional degeneracy, and lineage plasticity are factors that reduce drug efficacy and lead to acquired resistance. Recent studies have established the role of Hippo signaling network at the epicenter of cancer-associated transcriptional addiction and drug resistance. Thus, mechanistic understanding of cross talks between YAP/TAZ/TEAD with other TFs and chromatin remodelers may reveal gene regulatory networks that drive tumor invasion and growth. In this review, we focus on the nuclear effectors of YAP/TAZ that play a role in YAP-mediated transcriptional addictions and drug resistance in cancer (Table 1). Our primary focus is directed towards delineating these diverse pools of YAP/TAZ-associated transcriptional modules. For a more comprehensive view of Hippo pathway interactome, refer to [12,13]. We also discuss the involvement of phase separated biomolecular condensates in the Hippo signaling network that may play a critical role in cancer and therapeutic intervention. Finally, we cover future goals, as well as recent single-cell technological advances that can adequately dissect and explore resistance mechanisms and cancer addiction networks involving the Hippo signaling pathway.

## 2. TEADs

TEAD TFs (TEAD1-4) are evolutionarily conserved from *Drosophila* to humans. They regulate developmental processes transcending a wide variety of tissue types, from the formation of neural tubes to the development of heart, brain, and skeletal muscle. Several chromatin immunoprecipitation (ChIP) studies reveal that TEADs serve as the primary effectors of the YAP/TAZ signaling: (1) 78% of the TEAD4-bound promoters and enhancers are co-occupied by YAP/TAZ in NF2-null breast cancer cell line MDA-MB-231 [14]; (2) YAP and TEAD1 co-occupy >80% of the promoters in MCF10A breast cancer cells [15]; and (3) 86% of all the YAP1 peaks contain at least one TEAD-binding site in SF268 glioblastoma cell line [16].

TEADs bind to the DNA but are barely known to exert any transcriptional activity by themselves [17]. They form complexes with multiple TFs, coactivators, and chromatin remodelers to regulate gene expression in diseases and cancers. Despite the crucial role they play in tumorigenesis, the underlying molecular mechanism of TEAD-mediated transcriptional regulation is not well understood. TEADs share >80% homology in the DNA-binding domain and >70% homology in the cofactor-binding domain amongst themselves [18]. Despite having a high degree of sequence similarity, they control different sets of tissue-specific enhancer elements. The role of YAP/TAZ/TEAD as regulators of gene expression through distal enhancers and in cofactor switching has attracted significant interest in recent years, but the differential regulation of the individual TEADs and its significance in this emerging paradigm of transcriptional addiction in cancer largely remains unknown. They are also not known to harbor any oncogenic mutations.

## 3. YAP/TAZ/TEAD Control Gene Expression from Enhancers

Previously, it was thought that YAP/TAZ are coactivators for promoter driven transcription of a limited number of target genes [15]. However, recent technological advancement in the field of genome-wide ChIP seq and Transposase-Accessible Chromatin with high-throughput sequencing (ATAC-seq) have underscored the ability of YAP/TAZ/TEAD to tether to the chromatin at distal enhancer regions [14,16,19]. The importance of YAP/TAZ-mediated gene regulation through distant regulatory elements has shed light onto the global organization of relevant genes within the chromatin. TEADs are the primary mediators of YAP/TAZ recruitment to the chromatin. Interestingly, only 3.6% of the YAP/TAZ/TEAD binding is located in the promoter regions, in contrast to 91% of the complex located in the enhancer region, which can regulate expression of distal genes by chromatin looping [14].

ChIP data from various cell lines of different lineages and genetic backgrounds—such as glioblastoma line SF268 (YAP amplification), malignant mesothelioma line NCI-H2052 (NF2 mutation and LATS2 deletion), and non-transformed cells (IMR90)—has revealed that only a small set of YAP1/TEAD peaks are found within the gene transcription start sites (TSSs) while the majority are located at the distal active enhancers marked by histone post-translational modifications, including H3K27ac. The fact that YAP is required to maintain H3K27ac levels and TEAD occupancy at the YAP1-bound enhancer signifies its importance in establishing proper chromatin structures via positive feedback loop. Thus, YAP/TEAD distal binding is a general characteristic feature that tightly regulates transcription of a myriad of genes in both normal and cancer cells [16].

YAP further recruits mediator complexes to active enhancer sites, which can in turn regulate transcription via recruitment of Cyclin-Dependent Kinase 9 (CDK9), instilling “super enhancer”-like characteristics. Recruitment of CDK9 releases paused RNA Polymerase II (RNAPolII) at the promoters and induces elongation of YAP/TAZ targets to drive tumorigenesis in liver cancer cells. In the model liver cancer cell line HuCCT1, YAP occupies only 7% of the sites bound by TEADs, indicating that these super enhancer-like sites form a small subset of modules that are heavily overloaded with cofactors and activating histone marks. They play an important role in driving rapid gene expression in response to physiological and oncogenic cues and may play a role in YAP/TAZ-dependent cancer addiction as well [20].

In agreement with these studies, overexpression of the constitutively active form of YAP (YAP5SA) has been shown to reprogram cardiomyocytes into a fetal-like state by engaging a subset of global enhancers that are enriched for TEAD and AP-1 motifs. YAP overexpression promotes chromatin looping, which in turn increases enhancer–promoter contacts to access new target genes involved in developmental processes [21]. In flies, Yki can recruit mediator and histone methytranferase (HMT) complex via Ncoa6, a subunit of the Thritorax-related (Trr) methyltransferase complex to the promoter sites [22,23]. However, it is not known if Yki can engage in distal enhancer regulated gene expression. Ncoa6 can in turn regulate the expression of a subset of YAP targets in mammals, as reported in H69 cells. The regulation of enhancers by YAP/TAZ/TEAD is relevant to cancer as well. Recent studies reported enrichment of YAP/TAZ-bound enhancers in patient derived organoid models of human colorectal cancer. These enhanceromes are conserved across different patient-derived tumor types with a range of genetic alterations, serving as a common node where several oncogenic signaling converge to drive tumorigenesis [19].

It still remains a major challenge to correlate target gene expression with a specific set of enhancers. New technological advances in single cells can address this issue (covered in later section) and answer questions in this emerging field of enhancer deregulation in YAP driven cancer and their role in therapeutic resistance.

**Table 1 cells-11-02225-t001:** Nuclear effectors of YAP/TAZ/TEAD that play a role in YAP-mediated transcriptional addictions and drug resistance in cancers.

Factors	Conclusion	Tissue Origin	Reference
AP-1 and STAT	YAP/TAZ/TEAD and AP-1 transcription factors bind at the at the same genomic loci harboring TEAD and AP-1 composite sites. AP-1 enhances YAP/TAZ-induced oncogenic growth.	Breast	[14]
	TEAD and AP-1 co-occupy the cis-regulatory region. TEAD/AP-1 engages with steroid receptor c-activators 1-3 (SRC1-3) to regulate migration and invasion.	Brain, colon, lung, endometrium	[24]
	Vemurafenib (small-molecule inhibitor of BRAF V600E)-induced drug resistance is partially mediated by the activity of JUN and/or AP-1 and TEAD.	Skin	[25]
	AP-1 drives YAP-dependent transformations.	Skin, pancreas	[26,27]
	AP-1 is a transcriptional target of YAP/TAZ; induced AP-1 can collaborate with YAP/TAZ to promote organ growth.	Liver	[28]
	FOSL1/AP-1 acts as a common node in MAPK and Hippo pathways.	Colon and lung pharynx, esophagus, cervix, ovary	[29,30]
	YAP/TAZ are recruited by different forms of TEAD/STAT3/AP-1 complex depending on the cis-recruiting motifs to regulate different sets of YAP/TAZ target genes.	Breast	[31]
ERα/FOXA1	YAP/TEAD act as ERα cofactors to regulate ERα-bound enhancer activation by recruiting MED1.	Breast	[32]
BRD4	Enhancers occupied by YAP–TAZ show enrichment for BRD4, displaying super-enhancer-like characteristics and thus being sensitive to JQ1.	Breast	[33]
	ARID1A sequesters YAP/TAZ from binding to TEAD to decrease YAP/TAZ activity.	Liver	[34]
SWI/SNF	Pan-FGFR inhibition represses chromatin loading of BRG1, causing an epigenetic switch to promote YAP transcriptional dependency.	Breast	[35]
	Increased ACTL6A promotes loading of TEAD-YAP binding to BAF complexes, which can enhance co-binding of each other to the chromatin through a positive feedback loop.	Pharynx, lung, esophagus (squamous cells)	[36]

## 4. Role of AP-1 and STAT in YAP/TAZ/TEAD Transcriptional Regulation

Activator protein-1 (AP-1) is a well-characterized heterodimeric transcription factor, comprising two families of oncoproteins, namely, FOS and Jun. FOS family of proteins are early gene products and are rapidly induced in response to many cellular and extracellular cues, including cellular stress, developmental cues, and growth factors, as well as mitogens such as serum and lysophosphatidic acid (LPA) phorbol esters, which are also known to activate YAP activity [37]. The FOS family of proteins can regulate YAP activation and YAP/TAZ-derived phenotypes, such as cell survival, proliferation, and differentiation. Increasing evidence suggests that transcription control by AP-1 and YAP is highly interwoven and AP-1 and TEADs can synergistically drive tumor growth across several tumor types.

In the NF2-null breast cancer cell line (MDA-MB-231), 70% of the YAP/TAZ/TEAD-occupied enhancers also contained AP-1-binding motifs, making AP-1 the second most abundant motif after TEAD. Sequential ChIP seq analysis for YAP followed by JUN suggested that both TEAD and AP-1 can bind to the cis-regulatory elements bearing TEAD and AP-1 composite sites at the same time and can physically interact with each other. AP-1 synergizes with YAP to increase oncogenic growth in mammary cells via activating target genes that control S-phase entry and mitosis. AP-1 has elevated activity in skin tumorigenic induced by chemical carcinogenesis. YAP/TAZ-deficient mice failed to produce tumors when subjected to chemical carcinogenesis, underscoring the importance of YAP/TAZ in AP-1-mediated tumorigenesis [14].

Since YAP/TAZ rapidly translocate to the nucleus upon serum or LPA stimulation, they act as immediate sensors regulating early gene expression. Recent studies have shown that AP-1 is a direct transcriptional target of YAP/TAZ and is induced in response to mitogenic signals. Knocking out YAP in HEK293 cells severely reduces the expression of AP-1 upon LPA treatment, whereas overexpressing a constitutively active form of YAP (5SA-YAP) induces AP-1 expression under serum starvation. In addition, the induction of FOS expression is dependent on TEAD binding to YAP, which can then synergize with YAP/TAZ to drive target gene expression. A mutant version of YAP with defective TEAD binding (S94AYAP) fails to rescue AP-1 expression in YAP/TAZ KO cells. Consistent with these observations, inhibition of AP-1 significantly rescued YAP-mediated liver overgrowth in mice, highlighting a previously unknown role of AP-1/YAP/TAZ in the regulation of immediate early gene expression and organ growth [28]. There has also been evidence of YAP/AP-1 collaboration in pancreatic cancer progression. For example, deletion of LATS1/2 in organoids or in a mouse model of pancreatic cancer led to the activation of AP-1 and YAP target genes, where YAP physically interacts with the FOS/JUN complex. Concomitantly, treatment with AP-1 inhibitors reduces YAP-mediated transformations [27]. Interestingly, YAP is dispensable for normal epidermal homeostasis in comparison to basal cell carcinoma (BCCs). Mice containing a conditional KO of YAP have normal epidermis; however, YAP/TEAD promotes BCC growth by inducing AP-1 signaling. A loss in the level of YAP severely affected AP-1 family transcription factor c-JUN in BCCs, decreasing its stability and transcriptional activity.

YAP transcriptional regulation of AP-1 factors is also evolutionarily conserved in *Drosophila*. Activating transcription factor 3 (*Atf3)* is a direct transcriptional target of Sd, which is significantly upregulated in Ras driven tumor formation. Tumor specific gene expression in *Drosophila* is tightly regulated by a few key transcription factors, and AP-1 forms one of the major regulatory nodes. Loss of AP-1 or STATS can break this regulatory network by reducing the expression of tumor signature genes [38]. In the mammalian system, transcription of AP-1 is also affected by YAP as both TEAD and AP-1 bind to the promoter and enhancer sites, forming an autoregulatory loop. YAP impacts the transcription of known components of MAPK signaling in BCCs, but the upstream molecular mechanisms that connect YAP with the JNK-JUN axis are still not well understood [26]. Furthermore, it has been shown that TEAD and AP-1 co-occupy cis-regulatory region across a broad range of tumor cells such as the colon, lung, neuroblastoma, and endometrial cancer. TEAD/AP-1 transcription factors can engage with steroid receptor c-activators 1-3 (SRC1-3), which bridge between TEAD and AP-1 to regulate migration and invasion. SRC inhibition significantly inhibits the interaction of JUND with TEAD [24].

A recent study from our lab focused on an integrated strategy combining machine learning with chemicogenomics, to identify lineage-independent gene signatures for Hippo pathway in pan-cancer cell lines. We observed crosstalk between Hippo and MAPK signaling, converging at the level of AP-1 and serving as a common node to regulate gene expression. FOSL1 can interact with TEADs in the presence of YAP1. ATAC-seq under conditions of YAP depletion combined with MEK inhibition revealed decreased chromatin accessibility at TEAD and AP-1 binding sites [30]. Several studies show that YAP/TAZ activation serves as a bypass mechanism to overcome RAS/MAPK blockade. YAP1 rescues KRAS suppression in KRAS-dependent cancer cells. Tumors that escape KRAS suppression in a KRAS-driven murine lung cancer model show high YAP1 activity and upregulation of epithelial-to-mesenchymal transition (EMT)-like transcriptional programs. These gene signatures are jointly regulated by YAP1 and FOS, where YAP1 physically interacts with FOS at the promoters to drive YAP-induced transformation. YAP/TAZ activation also plays a crucial role in acquiring drug-induced resistance in BRAF and KRAS mutant cancer cells treated with EGFR/MAPK inhibitors. JUN and/or AP-1 and TEAD has been shown to induce resistance to vemurafenib (small-molecule inhibitor of BRAF V600E) in cultured patient-derived melanoma cells [25]. However, it still remains to be investigated whether AP-1-mediated cell cycle and EMT transcriptional programs might be one of the mechanisms to promote resistance to these other targeted therapies [29].

YAP/TAZ also act as a transcriptional coactivator for JUNB and STAT, promoting cellular transformation. YAP/TAZ, but not JUNB, are required for STAT3 phosphorylation during breast epithelial cell transformation. Along with TEAD/AP-1, YAP/TAZ target genes are also associated with STAT3, albeit to a lesser extent. YAP/TAZ are recruited by different forms of TEAD/STAT3/AP-1 complex depending on the cis-recruiting motifs, which can then regulate different sets of YAP/TAZ target genes. Interaction with these transcription factors is cell-type-dependent, causing specific gene expression profiles with distinct functions [31].

There is limited evidence on whether expression of YAP is regulated by other AP-1-like TFs. A recent study showed that YAP1 is a downstream effector of MAPK signaling activated by the FGFR axis, which promotes gastric cancer (GC) progression. JUN physically interacts with YAP1/TEAD4 complex and, consequently, knockdown of JUN impairs YAP1/TEAD4 complex formation, leading to reduced cell proliferation in model cell lines. FGFR activation promotes cJUN phosphorylation, and knockdown of FGFR significantly decreases YAP mRNA expression; however, the exact molecular mechanism is still unclear [39]. Further work is necessary to tease apart the synergistic roles from specialized roles of AP-1 and TEADs, as well as to assess the potential of AP-1 to be a therapeutic target for YAP/TAZ-dependent cancer. Moreover, it would be interesting to know whether this de novo induction of AP-1 expression is required for all or some of YAP/TAZ biological functions.

## 5. ERα/YAP/TEAD as a Downstream Effector of Hippo Signaling

Estrogen Receptor α (ERα) is a TF that controls cell proliferation and survival during tumor growth, and is elevated in 70% of breast cancer cases. The binding of estrogen promotes the localization of ERα from cytoplasm to nucleus, where it predominantly binds to distal enhancer regions to regulate target gene expression. ERα, in association with forkhead box A1 (FOXA1) is known to recruit other TFs that act as co-activators in-trans to form a stable enhancer activating machinery. Proximity labeling techniques in MCF7 breast cancer cell lines have identified YAP and TEAD4 as cofactors for ERα/FOXA1 on active estrogen-regulated enhancers. Instead of binding to its canonical cis-binding sites, TEAD4 can be recruited to the ERα enhancers in-trans to regulate expression of E2-induced ERα targets in addition to non-coding enhancer RNA (eRNA) transcription from estrogen-regulated enhancers. YAP1 and TEAD4 can also recruit mediator complex subunit 1 (MED1), an important enhancer activating machinery, to facilitate breast cancer cell growth [32]. This underscores an interplay between the Hippo pathway and ERα signaling pathway at the chromatin level, but the molecular mechanism that selectively recruits YAP/TEAD to active ERα enhancers is still unexplored.

Besides YAP/TAZ, LATS has also been implicated in the regulation of ER expression. In one study, LATS is required to maintain ER expression via YAP and TAZ inhibition. Contrary to its conventional role as a tumor suppressor, knockout of LATS inhibits growth of ERα+ breast cancer cells and mouse breast organoids by reducing the expression of ERα mRNA [40]. A different study reported that LATS1 overexpression targets ERα for ubiquitination Ddb1–cullin4-associated-factor 1 (DCAF1)-dependent proteasomal degradation independent of its kinase domain [41]. These two studies are contradictory, suggesting involvement of other YAP-independent mechanisms of ER expression in the presence of LATS. Further investigations are necessary to resolve the role of LATS/YAP in the regulation of ER signaling.

Recently, it has been shown that multiple TFs can interact with mediator subunits to form phase separated condensates. This interaction is facilitated by their activation domains in the TFs, which are intrinsically disordered, while their DNA-binding domains cooperate to anchor to their target sites. ER contains a C-terminal ligand dependent activation domain that facilitates formation of phase separated particles with MED1 upon estrogen stimulation [42]. Conformational changes induced by binding of other coactivators may also be important to recruit YAP/TEAD in-trans. Future studies necessitate the elucidation of molecular mechanisms that YAP/TAZ utilize to regulate ER gene expression and hormone resistance. Understanding such non-canonical roles of YAP/TEAD would open up exciting avenues for developing therapeutic strategies against endocrine-resistant breast cancer.

## 6. Role of BRD4 in Epigenetic Regulation of YAP/TAZ/TEAD-Mediated Transcription

Bromodomain-containing protein 4 (BRD4) is one of the well characterized histone acetyltransferases (HATs), belonging to the bromodomain and extra-terminal domain (BET) family. Recent studies have shown that YAP/TAZ can also contribute to transcriptional addiction by over engaging the chromatin reader BRD4 at YAP/TAZ/TEAD-bound promoters and enhancers and thereby exhibiting super enhancer-like properties.

In the breast cancer cell line (MDA-MB-231), YAP/TAZ recruit BRD4 to target enhancers and promoters to augment H3K122 acetylation, which is essential for loading of Pol II. YAP/TAZ targets are overloaded with BRD4 and are highly vulnerable to BET inhibitors. BET confers YAP/TAZ-mediated drug resistance in vemurafenib-treated BRAF mutant melanoma cells, which can be re-sensitized by JQ1 [33]. BRD4 and YAP are also known to induce profibrotic gene expressions in liver tumors [43]. YAP-induced liver tumorigenesis is suppressed in NF2 knockout mouse models using verteporfin (VP), which disrupts YAP–TEAD interaction [44]. Similarly, BRD4 inhibitor JQ1 has been shown to attenuate liver as well as lung fibrosis in murine models [45,46]. Thus, targeting YAP-TEAD and BRD4 in combination could be a potential therapeutic strategy for treating patients with organ fibrosis. Studies in lung cancer cells and esophageal adenocarcinoma cells (EAC) revealed that YAP, TAZ, and TEAD are transcriptional targets of BRD4 [47,48]. BRD4 also occupies the promoter of YAP1 to regulate the expression of YAP itself as well as expression of downstream target genes in EAC cells that have amplified YAP locus. Subjecting the cells to JQ1 treatment abrogated BRD4 binding to the YAP promoter. This highlights a feedforward role of BRD4 in boosting YAP/TAZ-dependent cell growth.

Apart from its role in transcription activation, BET can also suppress TAZ activity. Loss of function in LATS2, TAOK1, or NF2 in lung cancer cells results in nuclear accumulation of TAZ, leading to higher transcriptional activity of targets even in the presence of JQ1. According to this study, accumulation of TAZ is more likely to confer resistance to BET inhibitor than YAP. This supports the notion that YAP and TAZ serve tissue specific functions [49]. TAZ has been shown to be a novel mediator of Wnt/β-catenin and Hippo signaling. Wnt/β-catenin signaling releases TAZ from degradation complexes into the nucleus to regulate transcription. BRD4 inhibits Wnt signaling via suppression of β-catenin protein and thus upregulates TAZ and its downstream transcriptional activity but has no effect on YAP. This happens in a manner that is independent of transcriptional activity of BET proteins; however, the molecular mechanism by which it suppresses TAZ largely remains unknown [50]. This study has shown a potential mechanism of resistance to BET inhibitor, in lung cancer.

## 7. Role of SWI-SNF in Epigenetic Regulation of YAP/TAZ/TEAD-Mediated Transcription

Mammalian SWI/SNF (also known as BAF) complexes belong to the class of ATP-dependent chromatin remodelers, which contain a core ATPase that binds nucleosomes and disrupts DNA–histone interactions [51]. BAFs engage in dynamic complex formation with 11-15 subunits (ARID1A, ACTL6A discussed here) and one of the two mutually exclusive ATPases, BRG1 (SMARCA4) or BRM (SMARCA2). Together they recognize a diverse array of histone modifications, leading to both gene activation and repression under different cellular circumstances [52,53].

SWI/SNF complex has been shown to inhibit YAP-mediated transcription [34]. The ARID1A-containing SWI/SNF complex (ARID1A-SWI/SNF) specifically plays an important role in tumor suppression. It has been shown to modulate YAP/TAZ by directly interacting with it. Under low mechanical stress, ARID1A–SWI/SNF forms a complex with YAP/TAZ and sequesters YAP/TAZ from nuclear TEADs. Under high mechanical stress, as in tumorigenic conditions with stiff extracellular matrix, the ARID1A–SWI/SNF complex interacts with the nuclear F-actin and actin-related proteins (Arps), thus releasing YAP/TAZ to form complex with TEADs. Thus, the ARID1A–SWI/SNF-YAP/TAZ axis serves as a general sensor to cellular mechanotransduction. Genetic inactivation of ARID1A is widespread across several types of human cancer, and ARID1A-deficient tumor cells are very sensitive to robust epigenetic changes [34]. A recent study by Li et al. has reported the role of the SWI–SNF complex in developing resistance against FGFR inhibitor treatment in triple-negative breast cancers [35]. Although the protein level of YAP and phospho-YAP were not changed upon treatment of FGFR inhibitor, chromatin accessibility was increased at the YAP-dependent enhancer elements by loss of the BRG1–SWI/SNF complex. This further confirms that inhibition of SWI/SNF recruitment to the chromatin leads to epigenetic changes that drive YAP transcriptional dependency.

On the contrary, recruitment of ACTL6A to SWI/SNF (ACTL6A-BAF complexes) can increase chromatin accessibility at YAP/TEAD-bound enhancers. ACTL6A gene amplification plays a crucial role in squamous cell carcinoma (SCC) development by inducing an open chromatin landscape. ACTL6A reduction decreases TEAD accessibility, suggesting that it can regulate YAP/TAZ/TEAD-dependent oncogenic signaling. ACTL6A promotes TEAD-YAP binding to the chromatin, which can further recruit additional BAF complexes at the enhancer. This creates a positive feedback loop to maintain accessibility, and high transcriptional activity at TEAD-specific enhancer sites [36]. Consistent with this view, genetic studies in flies have shown that Brahma, the fly homolog of SWI/SNF ATPase, promotes transcriptional activity of Yki by binding with Sd and Yki at the promoter of Yki target genes [54]. In the mammalian system, AP-1 is also known to recruit the SWI/SNF (BAF) chromatin remodeling complex to initiate enhancer selection by establishing an open chromatin state [55]. Taken together, these studies highlight the cooperative role of SWI/SNF complex and YAP/TEAD transcriptional regulation in tumor progression and in development of resistance to targeted therapies.

## 8. Phase Separation

Biomolecular condensates are membrane-less subcellular bodies, formed by liquid-liquid phase separation (LLPS). They can regulate multiple intracellular processes including cooperative cellular events such as transcription initiation and elongation by compartmentalizing and concentrating known transcription hub components at super-enhancer-regulated genes. YAP/TAZ are known to physically recruit BRD4 and MED1 to super-enhancers. Interestingly, the intrinsically disordered regions (IDRs) of BRD4 and MED1 have been shown to facilitate formation of phase separated super-enhancer regions [56]. Multiple transcription factors can also interact with mediator subunits to form phase-separated condensates. This interaction is facilitated by the activation domains in the TFs that are intrinsically disordered, while their DNA-binding domains help to anchor it to target sites [42]. Recent studies have highlighted the role of phase separated YAP and TAZ in transcriptional regulation of the Hippo pathway. YAP forms phase-separated condensates under osmotic stress, whereas TAZ has an intrinsic tendency to form these condensates without any crowding in vitro. Phase separation of YAP is mediated by its transcriptional activation domain (TAD), which contains the IDRs. Formation of these condensates constitutes a dynamic process in which RNA Pol II is recruited to nuclear YAP condensates after stress induction. YAP condensates are also shown to be present in vivo; kidney medulla has higher number YAP-positive puncta compared to the cortical cells, owing to its high osmolarity [57]. TAZ condensate formations are driven primarily by their coiled coil (CC) domains, and partially by their WW domain. TAZ condensates compartmentalize other transcriptional machinery components such as BRD4, MED1, CDK9, and RNA Pol II to drive TAZ-specific gene expression [58]. Since YAP/TAZ are activated by many stress conditions, further work is necessary to address mechanistically how YAP/TAZ spatiotemporally orchestrates tissue-specific gene expression. DNA in situ hybridization followed by super resolution microscopy or new technological advances such as expansion microscopy would be useful to understand the inner dynamics of molecular organization within YAP/TAZ LLPS [59,60]. Furthermore, this property of YAP/TAZ can be leveraged to identify new drugs that target YAP/TAZ LLPS formation in disease pathology.

## 9. Outstanding Questions

In this review, we have extensively discussed how YAP/TAZ/TEAD cooperate with other master TFs and epigenetic cofactors to orchestrate a broad spectrum of transcriptional responses. This elaborate transcriptional network regulates the expression of key target genes in developmental and tumorigenic processes and is itself tightly modulated through feedback loops. Some pertinent questions remain: Are there additional master regulators that fuel transcriptional addiction in cancer? How is the context- and lineage-specific regulation of the Hippo pathway achieved? How does YAP/TAZ/TEAD specifically regulate gene expression under stress and external signaling cues? Is this specificity orchestrated through the recruitment of specific epigenetic and transcription factors to enhancer elements?

Our knowledge about the differential regulation of the four TEAD paralogs remains limited. Interaction of TEADs with their cofactors is highly context specific. Even though TEADs have redundant roles in certain developmental scenarios [61], very little is known about their unique roles in cancers. Future studies should focus on characterizing individual TEAD paralogs and their specific cofactors, understanding functional differences and similarities when they form complex with distinct set of cofactors, as well as their binding preferences for the cis-regulatory elements. A genome-wide chromatin mapping integrated with proteomics across different YAP-driven cancer types would be useful for uncovering novel regulators of YAP/TAZ-mediated transcriptional control networks, and this information can be leveraged for new therapeutic interventions. Understanding the heterogeneous molecular signatures and enhancer landscape of different tumors will advance the field of personalized medicine.

Given the importance of the Hippo pathway, there is tremendous interest in drugging YAP/TAZ/TEAD in oncology. Targeting YAP/TAZ still remains a major challenge, owing to the lack of structural rigidity and tolerability concerns. These challenges point towards drugging the YAP/TAZ transcriptional coactivators and TEADs. As reviewed elsewhere [10,62], there has been tremendous progress in the development of therapeutic approaches to disrupt YAP/TAZ–panTEAD interaction directly or by allosterically inhibiting TEAD palmitoylation. Since YAP/TAZ/TEADs also play a pivotal role in transcriptional addiction, targeting the transcriptional dependencies in YAP-driven cancers is another appealing strategy for therapeutic intervention. Drugs that target AP-1 activity or JUN/FOS stability have the potential to modulate YAP/TEAD activity indirectly. CDK9 inhibitor Flavopiridol and BET inhibitor JQ1 have also been used to inhibit YAP-driven tumor progression in cell lines and murine models [33,47]. Drugs inhibiting core components of transcriptional apparatus such as CDK7/9 and BET, or other epigenetic modulators are being evaluated. Small molecule screens to identify new molecules that specifically disrupt the binding of these cofactors with YAP/TEAD could be of interest as well.

High YAP/TAZ activity has also been implicated as a resistance mechanism to several therapeutic interventions against other oncogenic pathways. Drug-tolerant cancer cells can revert back to the sensitive state upon drug removal, or they undergo transcriptional and epigenetic reprogramming to become drug resistant cells. A pertinent question in the field will be to figure out how the downstream effectors of YAP/TAZ drive therapy resistance. Efforts should be made towards gaining insights into molecular mechanisms that cancer cells can exploit to activate bypass mechanisms and transcriptional reprogramming against small molecule inhibitors targeting TEAD or any of its cofactors. A comprehensive mechanistic understanding of transcriptional dysregulation in the context of tumor heterogeneity is imperative to developing effective combination therapeutic strategies.

## 10. Emerging Single-Cell Technologies and Future Perspectives

All the questions outlined above require the examination of transcriptional control in the context of cell identity and lineage plasticity. Given the highly heterogeneous and dynamic nature of tumor cells when exposed to treatment, traditional bulk assays may be insufficient to resolve resistance mechanisms that are present only in small subsets of cells. Therefore, there is a pressing need to apply the state-of-art, single-cell genomics technologies (Figure 3) to better understand the heterogeneity of Hippo signaling in drug response and resistance mechanisms, as well as lineage diversity in normal development.

Increasing evidence suggests that the Hippo pathway can have profound effects on specific cell populations in both normal developmental and tumorigenic processes. Applying single-cell ATAC-seq technologies, Little et al. discovered the specific localization of YAP/TAZ expression to the AT-1 cells of the lung and their role in recruiting lineage factor NKX2-1 to lineage-specific regions [63]. Subsequently, YAP/TAZ deletion shifts the cell fate of AT-1 cells towards the AT-2 lineage. Similarly in cancer, Castellan et al. performed single-cell RNA-seq of primary glioblastoma tumors and applied trajectory analysis to identify a population of glioblastoma stem-like cells (GSC) on the basis of their early pseudotime [64]. These GSC cells resemble early neuronal progenitors and are associated with poor clinical outcomes. Importantly, YAP and TAZ are predicted to be the master regulators of this GSC population—they are located at the top of the gene regulatory cascade and control downstream regulators such as FOXO1, FOS, and SALL1, which together regulate 73% of GSC signature genes. This study demonstrated the utility of single-cell RNA-seq studies in elucidating gene regulatory networks of Hippo signaling at high resolution across various lineages and rare tumor populations.

Single-cell RNA-seq alone may not be sufficient to assess changes in cell states. Increasing evidence has demonstrated that other modalities such as protein expression level, DNA accessibility, and histone/TF binding could dovetail nicely with transcriptome readout. These single-cell multiomic technologies can better resolve rare cell populations and provide mechanistic details of the cofactors associated with each lineage and/or each TEAD paralog. In terms of protein expression assay, CITE-seq is one of the earliest efforts to enable multiomic readout in single-cell genomics assays [65]. Using an oligo tagged antibody to label the protein, this method jointly measures the cell surface protein and gene expression. The follow-up technology called inCITE-seq tries to expand the application to intranuclear proteins [66].

As mentioned earlier, DNA accessibility is another important feature that provides unique insights into gene transcription regulation. Several methods have been developed to simultaneously detect the transcriptome and open chromatin information from the same cells, including sci-CAR [67], SNARE-seq [68], SHARE-seq [69], and the commercialized 10× Genomics Single Cell Multiome ATAC + Gene Expression assay. DNA accessibility conveys binding patterns for all the transcription factors; however, when studying the function of one or very few transcription factors, direct measurements of their interaction with the chromatin would be necessary. Recent advances in chromatin immunocleavage methods such as CUT&RUN [70] and CUT&TAG [71,72,73] have made it possible to survey such information at the single-cell level. Similarly, several efforts have been made to co-capture the TF–chromatin binding together with the transcriptome, such as Paired-Tag [74] and scCUT&TAG-pro [75].

In addition, a few groups have been trying to obtain all three modalities (RNA, open chromatin, and protein expression) in the same single-cell experiment, which resulted in newer technologies coupled to existing 10× Genomics Multiome system, such as NEAT-seq [76], TEA-seq [77], and ASAP-seq [78]. With ongoing advances in protocol optimization and analysis methods, we are hopeful that single-cell multiomics technologies can revolutionize our understanding of the Hippo pathway by delineating the lineage- and context- specific binding of individual TEAD paralogs, as well as their association with unique cofactors.

With the emergence of various modalities targeting TEADs, it is of high interest to compare the mechanisms of action amongst these pharmacological agents, as well as to understand how they may differ from genetic ablation of YAP/TAZ/TEAD. It is thus important to develop technologies to multiplex genetic and compound perturbation experiments. Traditionally, these experiments are limited by the numbers of perturbations and the types of readout in a single experiment. Several approaches have been introduced to overcome these issues by coupling single-cell genomics assays with the CRISPR-based genome editing technologies. In these assays, CRISPR is introduced to the samples in a pooled manner. Within every single cell, the sgRNA can be captured together with the transcriptome, which links the perturbation targets to their direct functional consequences. These methods include CRISP-seq [79], Perturb-seq [80,81], CROP-seq [82], and Mosaic-seq [83]. These single-cell CRISPR screen technologies can provide rich phenotypes for massive numbers of perturbations. A recent effort has expanded Perturb-seq to the genome-wide level for the first time [84]. Besides CRISPR-based gene perturbations, scRNA-seq can also be combined with the pooled ORF overexpression to enable screening for cell reprogramming [85,86] or coding variants of oncogenes [87]. Compound perturbation experiments, on the other hand, would require cells to be treated in the arrayed format rather than the pooled one. To achieve this goal, several multiplexing technologies have been developed, such as Cell Hashing [88], MULTI-seq [89], or split-and-pool scRNA-seq [90,91]. Follow-up studies have applied these technologies to study the compound response in different cancer cell lines [92,93]. Last but not least, several efforts are trying to combine the single-cell perturbation assays together with the multiomic readout we discussed in the previous section, such as ECCITE-seq [94] and Perturb-CITE-seq [95]. We hope that these highly multiplexed perturbation experiments can elucidate mode of action of various pharmacological agents and offer mechanistic insights into genetic perturbations across multiple cell populations and lineages.

## 11. Conclusions

We are hopeful that the era of single-cell technologies will usher in a granular understanding of the Hippo pathway—how YAP/TAZ/TEAD and master regulators cooperate to orchestrate lineage- and context-specific gene regulation, determine lineage fates, and confer treatment resistance. This finer resolution of regulatory control will ultimately enable us to design more precise therapeutic strategies, aimed at the right cellular populations.

## Figures and Tables

**Figure 1 cells-11-02225-f001:**
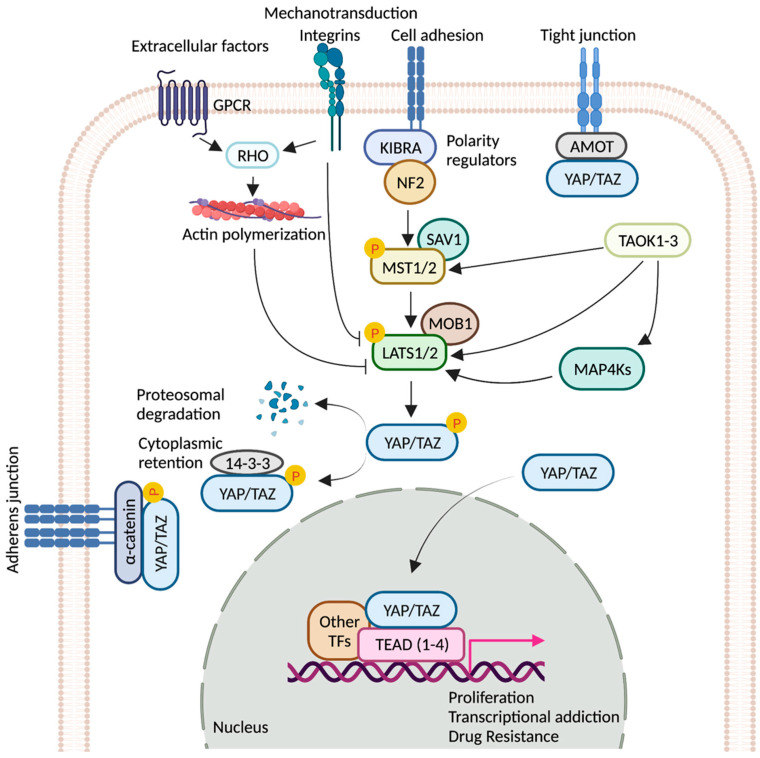
Regulation of YAP/TAZ activity by key signaling events. Schematic representation of the core components of the Hippo pathway. When the pathway is ON, a cascade of core kinases, composed of MST1/2 and LATS1/2, trigger phosphorylation of YAP/TAZ, which results in degradation or cytoplasmic retention of YAP/TAZ by 14-3-3. Various other signaling pathways and upstream effectors such as GPCRs (G protein-coupled receptors), TAOK family kinases, cell polarity, and adhesion regulators influence the activity of YAP/TAZ [4,5,6,7]. Mechanical cues relayed by extracellular-matrix-binding integrins and GPCR-mediated actin polymerization can inactivate the pathway. Unphosphorylated YAP/TAZ translocate into the nucleus, where it interacts with TEAD(1-4) and other cofactors. Together, they fuel the expression of pro-tumorigenic genes that can contribute to metastasis, transcriptional addiction, and drug resistance. Figure created with BioRender.com.

**Figure 2 cells-11-02225-f002:**
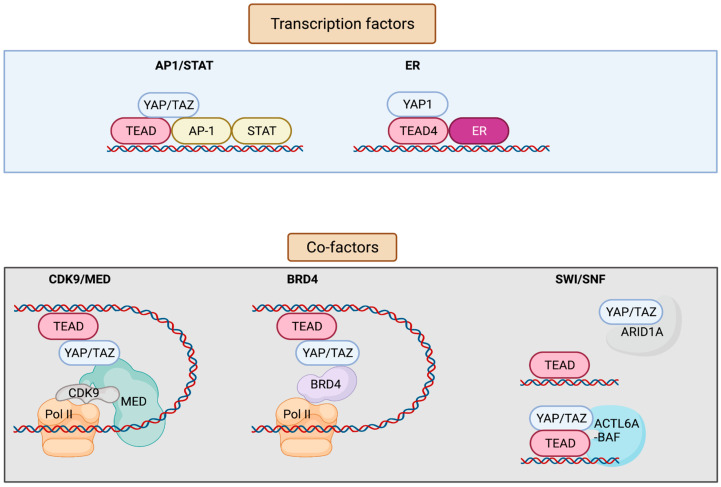
Interaction factors with YAP/TAZ/TEAD. YAP/TAZ/TEAD interact with transcription factors such as AP-1, STATs, and ER to drive transcription. YAP/TAZ/TEAD at enhancers recruit co-factors including Mediator and BRD4 that enable the release of paused Pol II and resumption of transcription elongation. YAP/TAZ/TEAD can also interact with various subunits of the SWI/SNF chromatin remodeling complex. ARID1A is thought to suppress YAP/TAZ transcriptional activity by sequestering YAP/TAZ from TEAD, whereas other subunits including ACTL6A and BRM are thought to promote YAP/TAZ transcriptional activity by enhancing chromatin accessibility at YAP/TAZ/TEAD bound sites. Figure created with BioRender.com.

**Figure 3 cells-11-02225-f003:**
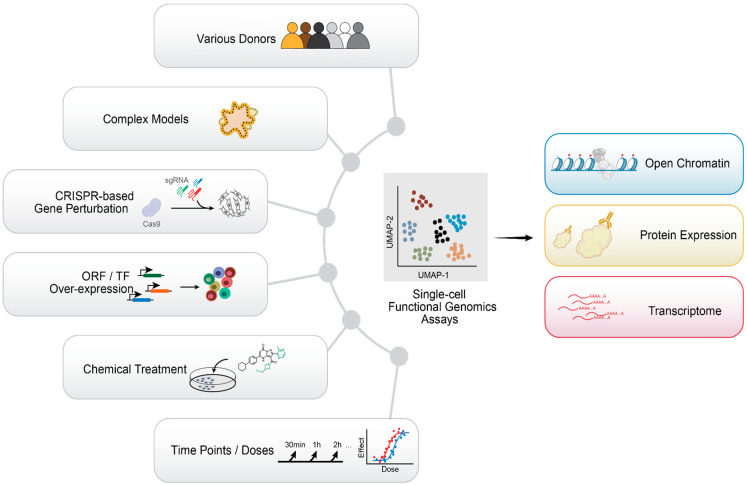
Single-cell technologies that can be applied to the studies of Hippo pathway. Recent progresses in single-cell genomics assays allow high content sample multiplexing (**left**) such as samples from different donors, complex model system with various cell types (i.e., organoids, tissue), genetic perturbations introduced by CRISPR system or ORF overexpression, and the compound perturbations at different time points or doses. As the output (**right**), different modalities such as RNA, protein, or chromatin accessibility can be simultaneously measured from the same cell.

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
