# Peer review of "Transcriptional Regulation of the Hippo Pathway: Current Understanding and Insights from Single-Cell Technologies"

_cells, 2022, doi:10.3390/cells11142225_

Round 1

Reviewer 1 Report

Sayantanee et al. submitted a very comprehensive, up-to-date and complete review of the Hippo pathway effectors YAP/TAZ in cooperation with other transcription factors and epigenetic cofactors that regulate gene transcription. The discussion is adequate and the presentation of the data is appropriate and well balanced. 

There are some minor points or suggestions that the reviewer wants to reminder the authors:

1. line 24 “PDZbinding” missed a space.

2. line 30 is confusing, is MST1/2 part of the NDR family kinase? And SAV1, MOB1 as adaptor proteins for the two kinases should be clearly stated.

3. how does YAP/TAZ/TEAD specifically regulate genes expression under different conditions, stress or in different tissues? Does it change the binding of other transcription factors and epigenetic cofactors to participate the regulation? If it is so, how do they switch the bindings in cells?

4. could you use a table to outline the different factors that bind to YAP/TAZ  and their functions? The reviewer thinks this may help the readers understand the review much easier.

5. could you also summaries the inhibitors that target yap/tead used in research and their limitations.

Author Response

Reviewer 1

Sayantanee et al. submitted a very comprehensive, up-to-date and complete review of the Hippo pathway effectors YAP/TAZ in cooperation with other transcription factors and epigenetic cofactors that regulate gene transcription. The discussion is adequate and the presentation of the data is appropriate and well balanced. 

We thank the reviewer for the positive feedback and the careful review of our manuscript. We have incorporated the suggestions as follows.

There are some minor points or suggestions that the reviewer wants to reminder the authors:

  1. line 24 “PDZbinding” missed a space.

We have made correction in line 23 in revised manuscript.

  1. line 30 is confusing, is MST1/2 part of the NDR family kinase? And SAV1, MOB1 as adaptor proteins for the two kinases should be clearly stated.

We thank the reviewer for pointing this out. We have made edits in the revised version from line 29-31.

  1. how does YAP/TAZ/TEAD specifically regulate genes expression under different conditions, stress or in different tissues? Does it change the binding of other transcription factors and epigenetic cofactors to participate the regulation? If it is so, how do they switch the bindings in cells?

We appreciate the reviewer for asking this question. The mechanism by which YAP/TAZ/TEAD regulates gene expression under different conditions, stress or in different tissues is not clearly understood yet and is an open area of research. We tried to address these points in the outstanding questions. We have made edits in the revised version, from line 439-441, to convey this idea more clearly.

  1. could you use a table to outline the different factors that bind to YAP/TAZ and their functions? The reviewer thinks this may help the readers understand the review much easier.

We thank the reviewer for this suggestion. We have now included Table 1 that summarizes the factors that bind to YAP/TAZ that is the focus of this review. We have also noted in the review that the table only includes factors that we have focused on and referenced other articles for a more comprehensive list in line 100-102.

  1. could you also summaries the inhibitors that target yap/tead used in research and their limitations.

We thank the reviewer for this suggestion. However, our review did not touch upon TEAD inhibitors because there is an extensive review on the different modalities of TEAD inhibition in the same series (Barry, Cells 2021, doi: https://doi.org/10.3390/cells10102715). We have cited this paper and others in line 463.

Reviewer 2 Report

This is a very nicely written review that summarized up-to-date understanding of transcriptome regulation by the Hippo pathway. 

I have the following suggestions for improving the accuracy of some conclusions in the review.

1. In Figure 1, mechanotransduction seems to regulate YAP/TAZ only through LATS1/2. However, Piccolo and other groups have also identified LATS1/2-independent mechano-regulation of YAP/TAZ. RhoA and actin cytoskeleton can play a role there.

2. TAOK kinase can also regulate MAP4Ks and potentially directly phosphorylate LATS1/2 (Dev Cell 2018 from Ip group and Mol Cell 2015 from Guan group).

3. Though there have been reports showing growth factor receptors may regulate MAP4K, the reviewer is not sure whether there is evidence showing that MAP4K regulation by RTK can also play a role in the Hippo pathway regulation. If so, please list the reference in the review. 

4. Previous studies showed that actin polymerization regulates LATS and YAP/TAZ likely independent of MST1/2. However, in Fig. 1, the arrow there is confusing. 

5. The conclusion that the authors made based on Ref 33 seems only putative. The actual experiments targeting YAP and BRD4 in organ fibrosis were not clearly performed. If so, please refer to original research articles instead of citing a review paper. 

6. The title contains 'single-cell'. However, a relatively small portion of the article discussed  Hippo pathway regulation and transcriptional output at the single-cell resolution. The title might be rephrased to ' from xxx to single cell'.

Author Response

Reviewer 2

This is a very nicely written review that summarized up-to-date understanding of transcriptome regulation by the Hippo pathway. 

We thank the reviewer for the positive feedback and careful review of our manuscript. We have incorporated the suggestions as follows.

I have the following suggestions for improving the accuracy of some conclusions in the review.

  1. In Figure 1, mechanotransduction seems to regulate YAP/TAZ only through LATS1/2. However, Piccolo and other groups have also identified LATS1/2-independent mechano-regulation of YAP/TAZ. RhoA and actin cytoskeleton can play a role there.

We thank the reviewer for this comment. We have updated figure1 and included some of the key ways of LATS1/2-independent mechano-regulation of YAP/TAZ.

TAOK kinase can also regulate MAP4Ks and potentially directly phosphorylate LATS1/2 (Dev Cell 2018 from Ip group and Mol Cell 2015 from Guan group).

We have added TAOK as an upstream regulator of MAPK4K in figure 1.

  1. Though there have been reports showing growth factor receptors may regulate MAP4K, the reviewer is not sure whether there is evidence showing that MAP4K regulation by RTK can also play a role in the Hippo pathway regulation. If so, please list the reference in the review. 

We appreciate the detailed feedback and have removed the regulation of MAP4K by RTK in the revised manuscript.

  1. Previous studies showed that actin polymerization regulates LATS and YAP/TAZ likely independent of MST1/2. However, in Fig. 1, the arrow there is confusing. 

We have modified Figure 1 to have actin polymerization directly pointing to LATS1/2.

  1. The conclusion that the authors made based on Ref 33 seems only putative. The actual experiments targeting YAP and BRD4 in organ fibrosis were not clearly performed. If so, please refer to original research articles instead of citing a review paper. 

We thank the reviewer for noting this. We have made edits in the revised version, from line 335-340.

  1. The title contains 'single-cell'. However, a relatively small portion of the article discussed Hippo pathway regulation and transcriptional output at the single-cell resolution. The title might be rephrased to ' from xxx to single cell'.

 We appreciate the reviewer's suggestion and have modified the title.

Reviewer 3 Report

Some minor suggestions:

1. Please be consistent: Hippo not hippo.

2. Hippo pathway is widely accepted as a tumor suppressor. Be careful and correct them when describe 'Hippo activity' throughout the MS. Be clear when use Hippo activity or YAP/TAZ activity.

Author Response

Reviewer 3

Some minor suggestions:

  1. Please be consistent: Hippo not hippo.

We apologize for the oversight and have made the corrections and ensured that the term “Hippo” is used consistently throughout the text.

  1. Hippo pathway is widely accepted as a tumor suppressor. Be careful and correct them when describe 'Hippo activity' throughout the MS. Be clear when use Hippo activity or YAP/TAZ activity.

We thank the reviewer for pointing this out. We ensured that YAP/TAZ activity is used when referring to oncogenic activity.